# Whole-Genome Analysis of Porcine Circovirus Type 2 in Russia

**DOI:** 10.3390/pathogens10121631

**Published:** 2021-12-16

**Authors:** Sergei Raev, Anton Yuzhakov, Taras Aliper

**Affiliations:** Federal State Budget Scientific Institution “Federal Scientific Center VIEV”, 109428 Moscow, Russia; anton_oskol@mail.ru (A.Y.); coronavirus@yandex.ru (T.A.)

**Keywords:** porcine circovirus type 2, genetic diversity, genotypes, phylogenetic analysis, PCV-2d

## Abstract

Porcine circovirus type 2 (PCV2) is the causative agent of porcine circovirus-associated diseases (PCVAD) that bring about significant economic losses in the pig industry all over the world. The aim of this study was to investigate the genetic diversity of PCV2 in Russia and characterize the available complete genome sequences. PCV2 DNA was detected at all investigated farms located in different regions of Russia. Whole-genome analysis demonstrated that the majority of PCV2 strains belonged to genotype PCV2d (12 out of 14), while PCV2a and PCV2b were only detected at 2 farms (one at each). Further analysis revealed that all antibody recognition sites in Russian PCV2 strains were different from the corresponding epitopes in a PCV2a vaccine strain, suggesting that PCV2a-based vaccines may only provide limited protection against these strains. PCV2d strains could be grouped into 3 distinct lines which shared 98.7–100% identity within open reading frame 2 (ORF2). It is the first study reporting the genetic diversity of PCV2 strains in Russia. Our data indicated that, similarly to China, Europe, and USA, PCV2a and PCV2b have largely been replaced by PCV2d.

## 1. Introduction

Porcine circovirus type 2 (PCV2) belongs to genus *Circovirus* in the *Circoviridae* family. Over 100 species are classified into *Circoviridae*, known to be one of the largest viral families. Members of this family have ssDNA genomes and demonstrate a relatively high mutation rate of 10^−3^–10^−4^ substitution/site/year. Along with PCV2, genus *Circovirus* includes three other porcine circoviruses: PCV1, PCV3 and PCV4. While PCV1 is nonpathogenic, pathogenicity of PCV3 and PCV4 remains to be evaluated [1,2]. Since the first diagnostic tool for PCV2 detection was created in the mid-1990s, several pathological conditions associated with PCV2 have been described, such as porcine dermatitis and nephropathy syndrome (PDNS), reproductive failure and proliferative and necrotizing pneumonia [3]. The role of PCV2 as the causative agent behind clinical disease was established after long debates, as demonstration of conformity with Koch’s postulates encountered some difficulty [4]. So far, porcine circovirus-associated diseases (PCVAD) are mostly acknowledged as an example of multifactorial disease [5].

PCV2 is a non-enveloped virus with a circular single-stranded DNA genome. Of all animal viruses, PCV2 has the smallest genome (1766 to 1768 nucleotides), comprising 11 open reading frames (ORF), at least 4 of which (ORF1-4) encode functional proteins [6]. ORF1 encodes two replication-related proteins Rep and Rep′ [7]. ORF2 encodes the viral capsid (Cap) protein, the major immunogenic protein implicated with immune response to PCV2 [8]. ORF3 encodes a protein involved in the induction of apoptosis [9]. Finally, a protein that is responsible for caspase activity suppression and regulation of CD4+ and CD8+ T lymphocytes in the course of PCV2 infection is encoded by ORF4 [10].

Initially observed to be genetically homogenous, PCV2 has gradually demonstrated increasing diversity over the last 30 years. In 2008, the first harmonized criteria for sub-species nomenclature were proposed, with nucleotide diversity cut-offs established as 3.5% and 2.0% for ORF2 sequences and complete genome sequences, respectively. The number of strain sequences available has since been growing, resulting in the identification of new genetically divergent clades. Criteria thus had to be reviewed, and new ones were offered for PCV2 classification: maximum intra-genotype p-distance of 13% (calculated based on ORF2 sequences), bootstrap support at the corresponding internal node over 70% and at least 15 available sequences. Among the 18 recognized clusters/genotypes, only 7 conform to these criteria. Only 3 clusters (3, 11 and 13) have been widely reported all over the world [5,11]. Up to early 2000s, PCV2a (cluster 3) remained the predominant genotype, after which the number of detected instances of PCV2b (cluster 11) gradually rose [12]. Since 2008, PCV2d (cluster 13) has been considered the most widespread genotype in Europe, China, and USA [13,14,15,16]. With the emergence of PCV2d, several PCVAD outbreaks at PCV-2a vaccinated farms have been reported, suggesting that available vaccines only provide partial protection against PCV2d [17,18,19]. Currently, the efficacy of PCV2a vaccines against already existing and newly emerged genotypes is given a lot of attention [10].

While the data on the genetic diversity of PCV2 in Europe, USA and China are ample and continually updated, similar data in Russia are scarce. The growing evidence that PCV2 proteins encoded by ORFs 1, 3 and 4 play a crucial role in PCV2 replication and pathogenesis emphasizes the advantages of full-length genome analysis as opposed to analysis based on ORF2 sequence.

The aim of the present study was to conduct phylogenetic analysis of PCV2 strains obtained in different geographical regions of Russia based on their complete genome sequences.

## 2. Results and Discussion

### 2.1. PCV2 Is Ubiquitous in Russia

PCV2 DNA was found at all 13 surveyed farms. Of the 125 samples from pigs that were tested, 57.6% were PCV2 DNA positive, while the overall rate of positive samples ranged from 13 to 100% at different farms (Figure 1). This is the first volume of evidence indicating that PCV2 is widely distributed in Russia. Despite the presence of PDNS-related clinical signs at one of the farms located in Kemerovo, viremia was only detected in 13.3% animals examined. This observation suggests that PCV2 viremia may not correlate with the manifestation of clinical signs even if they have been induced by PCV2. On the other hand, PCV2 is not the only causative agent of disorders such as respiratory distress syndrome and PDNS. Detection of PCV2 in organs should not be considered direct proof of PCV2 induced pathology either.

### 2.2. PCV-2d Is the Predominant Genotype in Russia

The phylogenetic tree was plotted using reference sequences to assess genetic relationships between previously known and 14 sequenced PCV2 strains (Figure 2). Sequence analysis showed that all 14 PCV2 genomes were 1757 or 1767 bp in length, while the ORF2 nucleotide sequences measured 702 or 705 bp. Pairwise comparison analysis revealed that the identity of most genomes’ complete nucleotide sequences ranged from 96% and 100%, with only one exception (Belgorod RA18) that only demonstrated 93.7% nucleotide identity with other sequences Phylogenetic analysis indicated that 14 strains could be placed into three genotypes/clusters: PCV2a (1 out of 14), PCV2b (1 out of 14) and PCV2d (12 out of 14). This data corroborated previous findings that PCV2d is currently the predominant genotype of PCV2 in most countries where pork production is well-developed [13,14,20]. Remarkably, our data suggested that in the Siberian and Far Eastern districts, which border on the People’s Republic of China, PCV2a and PCV2b have been completely replaced by PCV2d [21].

### 2.3. The Antibody Recognition Domains Sequences in ORF-2 Gene Are Affected to a Significant Extent in Russian PCV-2 Strains

There are several epitopes, including virus-neutralizing ones, which have been recognized in the capsid protein [5]. Even single amino acid mutations within ORF2 have been shown to reduce virus neutralization [22]. A comparison of the capsid protein sequences between the vaccine strain [23] with Russian strains revealed that all four known antibody recognition domains (A, B, C and D) [24] carried amino acid substitutions (Figure 3). Remarkably, these substitutions were most noticeable in PCV2d rather than in PCV2a and PCV2b. The presence of common epitopes and cellular cross-immunity between PCV2a and PCV2b have demonstrated high efficacy of PCV2a-based vaccines against these genotypes. On the other hand, a lack of clear evidence on the efficacy of currently available vaccines against the globally prevalent PCV2d genotype and the ability of PCV2d to replicate under vaccine immunity pressure remain a serious concern for the pig industry. Development of a novel PCV2d or PCV2a/PCV2d vaccine might be considered as an option to induce broad protection against currently circulating PCV2 genotypes.

### 2.4. Belgorod RA18 (PCV2a) Has a Unique Deletion in the Origin of Replication

An 11-nucleotide deletion in origin of replication (Ori) was observed in RA18 strain (Figure 4). Ori consists of a stem-loop structure and four hexamer sequences (H1-4), which are essential for the initiation and termination of PCV2 replication. H1/2 play a critical role in the initiation of PCV2 replication, while H3 and H4 are optional binding sites for the Rep protein. Thus, our finding is the first piece of evidence suggesting that the presence of H4 is not essential for virus replication [25].

### 2.5. PCV-2d Strains Can Be Classified into 3 Lines Indistinguishable on the Basis of ORF2 Sequences

PCV-2d strains shared 98.2% to 100% nucleotide identity. Further analysis of PCV-2d whole-genome sequences revealed that they could be classified into 3 distinct lines, with the bootstrap exceeding 80%. As shown in Figure 2a, the first line, represented by 2 strains (Belgorod M 18 and Kurskaya 2020), had 2 unique amino acid substitutions within CAP (R169G) and REP (I83V) proteins compared to the second line. Both first- and second-line strains (strains Kemerovo_Ch_522 and Kemerovo_Sl_809 sharing a common border) contained amino acid substitutions in REP (N6S and L77F) and ORF3 (T29A, G41R, Q89E and N103Y) sequences (Figure 5). The third line is most closely related to the PCV2a genotype. Thus, while all PCV-2d strains formed three distinct lines, they were not found distinguishable based on Cap gene sequences, suggesting that all these strains may be regarded as immunologically homogenous.

Phylogenetic analysis, based on ORF2, is known to produce similar results to whole viral genome analysis [26], making the ORF2 gene more suitable and thus widely used for epidemiological research into PCV2. Our data indicated that whole-genome sequence analysis allowed evaluating genetic diversity even among immunologically homogenous PCV2 strains.

## 3. Materials and Methods

### 3.1. Farms

Samples were collected from 2018 to 2020 at large industrial pig farms located in the Central (6), Far Eastern (1), Northwestern (1), Siberian (4), and Ural (1) federal districts of Russia. At all farms vaccination against PCV2 had been conducted for more than 3 years before this investigation. Despite vaccination, PCVAD, including PDNS, were observed at some farms.

### 3.2. Sample Collection, DNA Extraction

Serum samples were taken from live pigs and lung/lymph nodes were collected during necropsy. Lung and lymph nodes from each animal were pulled and used to tissue homogenates prepare. Homogenized tissue (lungs, lymph nodes) and blood serum (Appendix A) were used for DNA extraction. DNA extraction and detection of PCV2 were performed using a commercial PCR test-kit (Vetbiochem, Russia).

### 3.3. Whole-Genome Sequencing

Sequencing primers (Table 1) were designed based on PCV2 sequences deposited in GenBank. The resulting overlapping PCR fragments covered the entire genome of the virus. PCR was performed using the Acculong Polymerase kit (Alfa-ferment, Moscow, Russia). The PCR program was as follows: (1) 94 °C for 5 min; (2) 30 cycles of: 94 °C for 20 s, Ta °C for 20 s, 68 °C for 1 min; (3) 68 °C for 10 min. Amplified fragments were analyzed by way of 1% agarose gel electrophoresis. The amplified fragments were isolated from gels using the Silica Bead DNA Gel Extraction Kit (Thermo Fisher Scientific, Waltham, MA, USA). The PCR products were sequenced using the Big Dye Terminator Cycle Sequencing Kit (Applied Biosystems, Waltham, MA, USA) and an ABI PRISM 3130 Genetic Analyzer (Applied Biosystems, Waltham, MA, USA).

### 3.4. Genome Alignment and Phylogenetic Analysis

Nucleotide sequences were analyzed using Lasergene 11.1.0. (DNASTAR, Madison, WI, USA). Multiple alignment was performed using ClustalW (BioEdit 7.2) and MUSCLE (MEGA 7.0.18). Phylogenetic dendrograms were inferred using the maximum likelihood method, GTR model (MEGA 7.0.18). The topology of the trees was confirmed over 1000 bootstrap replication steps [27]. Sequences were compared to those available in the GenBank database using the Standard Nucleotide BLAST software package (http://www.ncbi.nlm.nih.gov/BLAST, accessed on 4 October 2021).

## 4. Conclusions

The present study is the first work reporting whole-genome sequence analysis of PCV2 strains circulating in Russia. Our data indicated that PCV2d is widely distributed in Russia. Whole-genome sequence analysis of PCV2d strains revealed that even without considerable substitutions/mutations in ORF2 gene, these viruses clustered as three separated lines within the PCV2d genotype.

## Figures and Tables

**Figure 1 pathogens-10-01631-f001:**
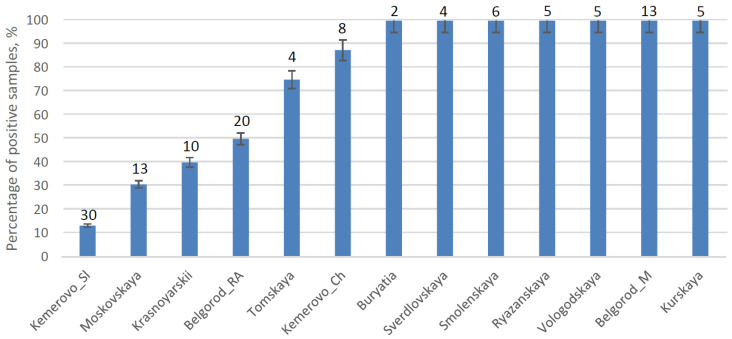
Percentage of PCV2 DNA positive samples collected from swine farms. The number of samples used for each farm is shown above the bar.

**Figure 2 pathogens-10-01631-f002:**
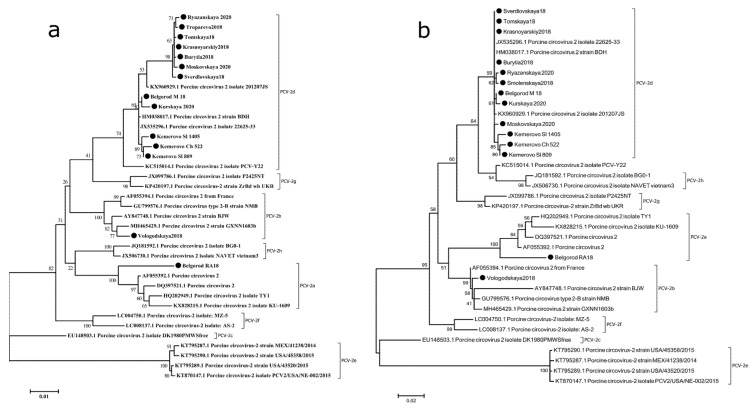
Phylogenetic trees of complete genome nucleotide sequences (**a**) and ORF2 nucleotide sequences (**b**) of PCV-2 strains. Multiple sequence alignment was performed using the ClustalW method. Bootstrap confidence limits are shown at each node. The strains identified in this study are indicated by circles (●).

**Figure 3 pathogens-10-01631-f003:**
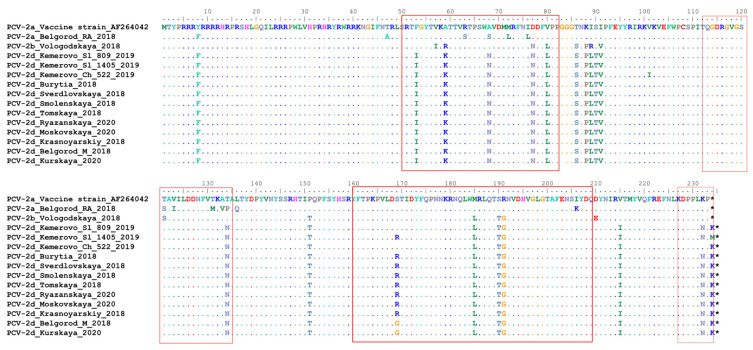
Complete alignment of CAP protein amino acid sequences of Russian PCV-2 strains. Multiple sequence alignment was performed using the Muscle method. The top line corresponds to the PCV-2a strain present in a commercial vaccine. Dots and hyphens represent identical amino acid positions and gapped positions, respectively. Asterisks represent stop codons Antibody recognition domains (A (51–81), B (113–134), C (161–208) and D (228–233)) are shown in red boxes.

**Figure 4 pathogens-10-01631-f004:**
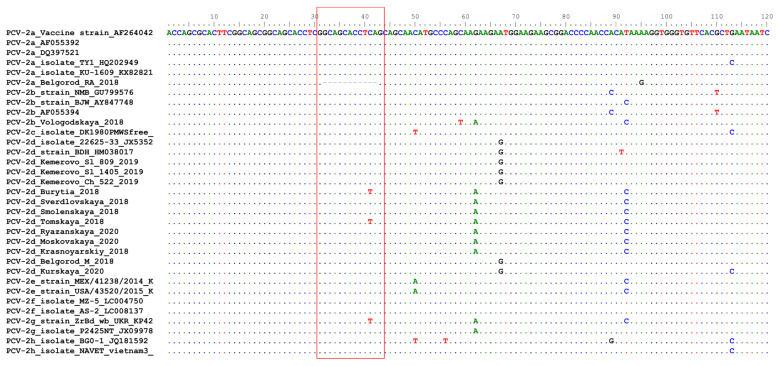
Partial alignment of Ori nucleotide sequences from Russian and reference PCV2 strains. Dots and hyphens represent identical amino acid positions and gapped positions, respectively. The fragment with an 11-nucleotide deletion in strain Belgorod_RA18 and corresponding fragments in other viruses are shown in a red box.

**Figure 5 pathogens-10-01631-f005:**
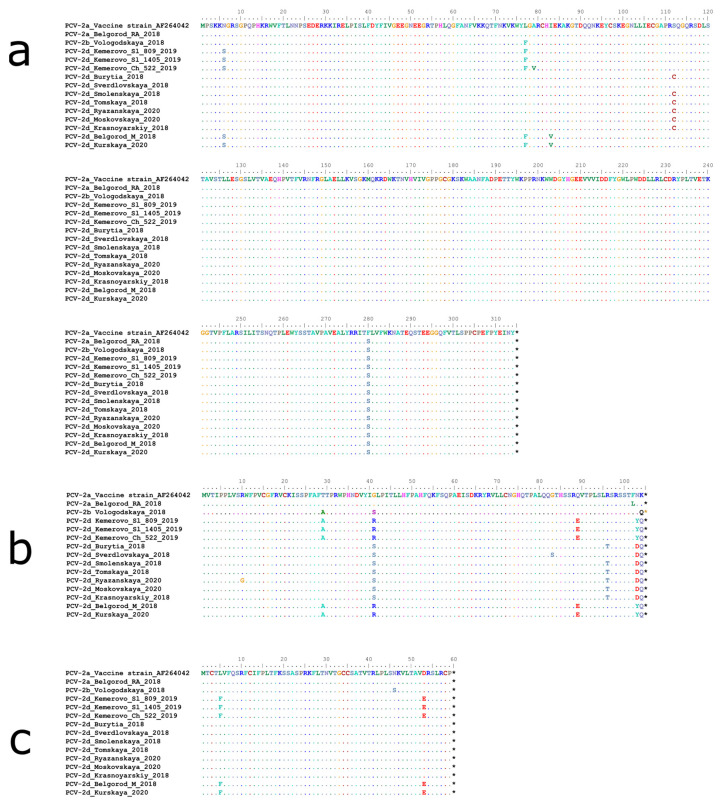
Complete alignment of nonstructural proteins’ amino acid sequences of Russian PCV-2 strains: (**a**) REP, (**b**) ORF3 and (**c**) ORF4. Multiple sequence alignment was performed using the Muscle method. The top line corresponds to the sequence of a PCV-2a strain used in a commercial vaccine. Dots and hyphens represent identical amino acid positions and gapped positions, respectively. Asterisks represent stop codons.

**Table 1 pathogens-10-01631-t001:** List of oligonucleotide primers used for the amplification of the PCV2 genome.

Primer Pair	Primer Location (Numbers Correspond to Position within the PCV2 Genome)	Primer Sequence (5′-3′)	Amplicon Size, nt	Ta, °C
PCV2_1F	229	GGTTCGCTAATTTTGTGAAGA	596	56
PCV2_1R	824	GGTCTGATTGCTGGTAATCA		
PCV2_2F	686	TGTTATTGATGACTTTTATG	641	48
PCV2_2R	1326	TATGTAAACTACTCCTCCC		
PCV2_3F	1073	TTAAATTCTCTAAATTGTAC	470	46
PCV2_3R	1542	GTGGACATGATGAGATT		
PCV2_4F	1443	TAACCTTTCTTATTCTGTA	683	46
PCV2_4R	359	TAAGTTGCCTTCTTTACT		

## Data Availability

Not applicable.

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
