# Peer review of "Whole-Genome Analysis of Porcine Circovirus Type 2 in Russia"

_pathogens, 2021, doi:10.3390/pathogens10121631_

Round 1

Reviewer 1 Report

The authors report the genetic analysis of PCV2 from samples collected in various regions of Russia. Although not extremely novel, the presented data contribute to the understanding of the PCV2 genetic variation in an important pig producer country. Besides a complete English review for grammatical, cohesion and coherence corrections and improvement of the data presentation in figure 1, I consider the manuscript proper as "communication".

Author Response

Reviewer 1.

Comments and Suggestions for Authors

The authors report the genetic analysis of PCV2 from samples collected in various regions of Russia. Although not extremely novel, the presented data contribute to the understanding of the PCV2 genetic variation in an important pig producer country. Besides a complete English review for grammatical, cohesion and coherence corrections and improvement of the data presentation in figure 1, I consider the manuscript proper as "communication".

Answer: We would like to express our sincere gratitude to the reviewer for his time. Our manuscript type is “Communication”

Reviewer 2 Report

Thank you very much for the opportunity to review.

General comment

The study reported Whole-Genome Analysis of Porcine Circovirus Type 2 in Russia

In general, the paper was well written and understandable. An interesting approach is reported, unique and novelty can be considered as described in the article.

Specific comments

1.It would be worthwhile to write the economic damage caused by PCV2 in Russia

2. Have they found an association between different genotypes and clinical diseases, economic damage?

Author Response

Reviewer 2.

Comments and Suggestions for Authors

Thank you very much for the opportunity to review.

General comment

The study reported Whole-Genome Analysis of Porcine Circovirus Type 2 in Russia

In general, the paper was well written and understandable. An interesting approach is reported, unique and novelty can be considered as described in the article.

Answer: We would like to express our sincere gratitude to the reviewer for his time

Specific comments

1.It would be worthwhile to write the economic damage caused by PCV2 in Russia

Answer: Unfortunately, we have no data on the economic damage from circovirus diseases in Russia.

  1. Have they found an association between different genotypes and clinical diseases, economic damage?

Answer: Due to the limited data about health conditions in farms investigated, we were not able to find any association between different genotypes and clinical diseases.

Reviewer 3 Report

Whole-Genome Analysis of Porcine Circovirus Type 2 in Russia

The aim of reviewed manuscript was to conduct the whole-genome phylogenetic analysis of PCV2 isolates obtained from commercial pig farm situated in different geographical regions of Russia.

Generally the subject of the manuscript is interesting and valid. However the manuscript requires a major improvement in the English language and in same part of Material and methods and of Resulat section.

Materials and methods used in the study are relatively adequately described. However I could not find in supplementary file the descriptive data about the number of samples collected per farm, number of different type of sample, and age category of pigs from whom the samples were collected. Moreover the information where the samples (lymph node, lungs, serum) were collected should be added (slaughterhouse ?). In addition information in which farm and what kind of the clinical symptoms of PCVAD was observed should be added.

In Results section Figure 1 should be thoroughly changed. Type of sample should be added (serum?/lungs?) It would be better to prepare Table including type of analysed sample and percentage of positive samples (serum/lungs/lymph node). Beside the percentage of positive samples a confidence interval (CI) should be added.

In addition, the section ‘‘2.1. PCV2 is ubiquitous in Russia’’ need to be rewritten. There data about percentage of positive lungs and /or lymph node is missing.

There is only description about the results of Kemerovo farm “Despite on the presence of PDNS-related clinical signs in a Kemerovo farm, viremia was detected in only 13.3% animals examined”. Have PDNS symptoms been observed only in Kemerovo farm ? Data about clinical symptoms of PCVAD in other farm should be added, and correlation with viremia as well.  

The conclusion is generally well described and comprehensive.

Line 23: Word “virus” should be removed

Author Response

Reviewer 3.

Comments and Suggestions for Authors

Whole-Genome Analysis of Porcine Circovirus Type 2 in Russia

The aim of reviewed manuscript was to conduct the whole-genome phylogenetic analysis of PCV2 isolates obtained from commercial pig farm situated in different geographical regions of Russia.

Generally, the subject of the manuscript is interesting and valid. However, the manuscript requires a major improvement in the English language and in same part of Material and methods and of Result section.

Answer: We would like to express our sincere gratitude to the reviewer for his time and valuable expertise that we used to improve several gaps in the manuscript. We have addressed point by point every issue mentioned by reviewer and made changes in the manuscript accordingly.

Materials and methods used in the study are relatively adequately described. However, I could not find in supplementary file the descriptive data about the number of samples collected per farm, number of different types of samples, and age category of pigs from whom the samples were collected 

Answer: All of this data is presented in the Table 1. Supplementary, columns “Number of samples used”, “Sample” and “Age”, respectively.

Moreover, the information where the samples (lymph node, lungs, serum) were collected should be added (slaughterhouse ?).

Answer: The following information has been added in Materials and Methods chapter: Serum samples were taken from live piglets and lung/lymph nodes were collected during necropsy.

In addition, information in which farm and what kind of the clinical symptoms of PCVAD was observed should be added.

Answer: This data is presented in the “Health status” column of Table 1. Supplementary.

In Results section Figure 1 should be thoroughly changed. Type of sample should be added (serum?/lungs?) It would be better to prepare Table including type of analyzed sample and percentage of positive samples (serum/lungs/lymph node).

Answer: we changed Materials and Methods section. We didn’t sampled lung and lymph nodes separately. “ Serum samples were taken from live piglets and lung/lymph nodes were collected during necropsy. Lung and lymph nodes from each animal were pulled and used to tissue homogenates prepare. Tissue homogenates (lungs, lymph nodes) and blood serum (Supplementary Table) were used for DNA extraction.”

Beside the percentage of positive samples, a confidence interval (CI) should be added. In addition, the section ‘‘2.1. PCV2 is ubiquitous in Russia’’ need to be rewritten. There data about percentage of positive lungs and /or lymph node is missing. There is only description about the results of Kemerovo farm “Despite on the presence of PDNS-related clinical signs in a Kemerovo farm, viremia was detected in only 13.3% animals examined”. Have PDNS symptoms been observed only in Kemerovo farm ? Data about clinical symptoms of PCVAD in other farm should be added, and correlation with viremia as well. 

Answer: All text was rewritten. This data is presented in the “Health status” column of Table 1. Supplementary.

As mentioned in Table 1. Supplementary, PDNS were observed at…. Serums samples were collected at Kemerovo farms only.

Answer: unfortunately we were unable to select other material on this farm

The conclusion is generally well described and comprehensive.

Answer: thank you

Line 23: Word “virus” should be removed

Answer: Page 1 Line 23 have been updated as recommended.

Round 2

Reviewer 3 Report

Comments and Suggestions for Authors

Generally , the authors made the recommended corrections and the revised version of the manuscript may be published in Pathogens.

I have only one minor comments regarding the Figure  1. - the y axis description is illegible and needs improvement. Also beside the percentage of positive samples, a confidence interval (CI) should be added in the bars in Figure 1.

Author Response

Generally , the authors made the recommended corrections and the revised version of the manuscript may be published in Pathogens.

Answer: We would like to express our sincere gratitude to the reviewer for his time and valuable expertise that we used to improve several gaps in the manuscript. We have addressed point by point every issue mentioned by reviewer and made changes in the manuscript accordingly.

I have only one minor comments regarding the Figure  1. - the y axis description is illegible and needs improvement. Also beside the percentage of positive samples, a confidence interval (CI) should be added in the bars in Figure 1.

Answer: Figure 1 has been updated as recommended